# Measuring Respirable Crystalline Silica (Quartz) from Powdery Materials through Sedimentation and X-ray Diffractometry

**DOI:** 10.3390/toxics12040241

**Published:** 2024-03-25

**Authors:** Tapani Tuomi, Jussi Lyyränen

**Affiliations:** Finnish Institute of Occupational Health, Topeliuksenkatu 41 B, Työterveyslaitos, FI-00032 Helsinki, Finland; jussi.lyyranen@ttl.fi

**Keywords:** respirable crystalline silica, construction materials, building products, EU Regulation No. 1272/2008, quartz exposure

## Abstract

When possible, choosing materials with a low quartz content is the most effective and cost-efficient way to prevent the respirable quartz exposure of workers and other end users of powdery products. Therefore, methods are needed to analyze low amounts of quartz from powdery products, such as sand, gravel, plaster, cement, and concrete. To this end, we present a method to analyze respirable dust and quartz from powdered materials, such as construction products. The method includes separation of the respirable dust fraction by liquid sedimentation, followed by gravimetric analysis and determination of the crystalline silica content by X-ray diffractometry. While also aiding in the development of less harmful products, analysis of the quartz concentration of powdery products is statutory in Eu countries, excluding natural products not chemically modified. According to EU Regulation No. 1272/2008, products must be classified if they contain harmful substances in concentrations above 0.1 wt.%, and clauses pertaining to cancerous properties and harmfulness to lungs should be included. Also, mineral producers in the EU recommend that products containing respirable quartz should be labelled based on their quartz concentration, provided the concentration exceeds 1 wt.%. The present method meets these needs. The analysis can be performed in parallel from 50 to 1000 mg (dry weight) of powdery materials. The quantitative limit of determination was 10 µg per sample, corresponding to 0.01 wt.%, and the linear range 0.02–10 wt.% (10–5000 µg quartz per sample, Pearson correlation coefficient 0.99). The accuracy of the method was 82% and the repeatability 11%.

## 1. Introduction

Crystalline silica is a common ingredient in building products and other materials containing or composed of stone, gravel, clay, or sand. Such materials include cement, concrete, asphalt, bricks, plasters, tiles, and slates. Exposure to respirable silica during extended periods or during short periods of high exposure causes silicosis, and it may also lead to the development of lung cancer [1]. Workers exposed to, on average, 0.05 mg/m^3^ of respirable silica during their working career stand at high risk of contracting both silicosis and lung cancer. For such workers, the risk ratio for death from silicosis has been estimated to be 0.006 and the corresponding ratio for lung cancer 0.019 [2,3]. In fact, the risk of lung cancer death from 5 years of work in the construction industry has recently been estimated to be comparable with the risk from a personal cancer history or a family history of cancer, with the most frequent cancerous agents being asbestos and respirable quartz, respectively [4]. Respirable silica is released into the air when using any of the dusty materials mentioned or when machining solid materials containing crystalline silica. In addition to construction work, occupations of potentially high exposure to crystalline silica and where silicosis has been reported include miners and quarrymen, foundry workers, stonemasons, glass and ceramics kilnmen, and stone cutters [5,6,7]. According to recent estimates, approximately two-thirds of all workers exposed to quartz in Finland are building workers [6]. Consequently, building workers are overrepresented among workers contracting silicosis.

It follows that measures lowering quartz exposure in construction work tasks are much needed and will help in lowering the overall incidence of diseases related to quartz exposure. When possible, the most effective means to control exposure in construction or production work tasks is either excluding the source of exposure altogether or, when this is not feasible, minimizing emissions at their source. Consequently, when using dry powdered or granulated materials, one should choose materials that are as low as possible in their respirable dust and quartz contents. Moreover, when choosing tiles, kitchen countertops, and other materials that will require cutting or drilling during their installation, it is better to choose a material free of quartz or with as a low quartz content as possible.

In EU countries, crystalline silica and other category 1 carcinogens placed on the market, either as such or as ingredients in mixtures, are subject to the classification obligation under, unless they are present in quantities below 0.1% (*w*/*w*) [8]. It should also be considered that respirable quartz causes silicosis (harmful to organs). Consequently, such products should include the warning “May cause lung cancer by inhalation” and “Causes damage to lungs”. Minerals, ores, gravel, sand, and other natural products containing quartz are not covered by the REACH regulation (Registration, Evaluation, Authorisation, and restriction of Chemicals, 1907/2006), if they are not chemically modified [9]. However, industrial mineral producers in the EU (IMA) have, based on a Hazard Assessment, jointly determined it appropriate to classify even these non-modified products containing crystalline silicas (fine fraction) based on their crystalline silica content [10]. Consequently, such products should include the labelling STOT RE 1 (Specific Target Organ Toxicity upon Repeated Exposure, Category 1, H372), if the respirable crystalline silica concentration is equal to or greater than 10 wt.%. And if the respirable crystalline silica concentration is between 1.0 and 10 wt.%, the labelling should be STOT RE 2 (Specific Target Organ Toxicity upon Repeated Exposure, Category 2, H373). If the respirable crystalline silica (fine fraction) content in mixtures and substances is below 1.0 wt.%, no classification is required according to IMA.

Given these considerations, accurate methods are needed to determine the content of respirable quartz and total quartz in, for instance, construction materials. Respirable quartz should be determined in products that may become partially airborne during their handling and use. Total quartz content, regardless of particle size, is of interest when choosing materials that during their span of use are potentially machined in a manner that enables the formation and spreading of airborne respirable quartz.

To promote the well-being of quartz-exposed workers and reduce exposure-related harms, industries where quartz exposure is a concern in the EU, excluding the construction industry, signed “The Agreement on Workers Health Protection through the Good Handling and Use of Crystalline Silica and Products Containing it” in 2006 (NEPSI treaty) [11]. According to the treaty, as well as all major standards and reference methods of research institutes, quartz should be determined by using either X-ray diffraction (XRD) or Fourier-Transform Infrared Spectroscopy (FTIR), upon first extracting the desired particle size range according to European Standard EN 481 [11,12]. The respirable fraction is the fraction of inhalable particles that reach the alveolar region of the lung and is described in standard EN 481 by a cumulative log-normal distribution, with a median of 4.25 μm and a geometric standard deviation of 1.5 [12]. According to EN 481, the respirable fraction, i.e., the percentage of the inhalable fraction which is to be included at any given aerodynamic diameter, shall be given by named distribution and varies from 1.3% at 10 µm (0% at 16 µm) to 97% at 1 µm, with a so-called 50% cut-off at 4 µm. Airborne respirable dust according to the EN 481 convention is commonly extracted by using cyclones, with the air flow calibrated to correspond to a 50% cut-off at 4 µm [11]. However, withdrawing airborne dust reproducibly from powdered products is challenging with the methods presently available. For instance, the rotating drum method, used to analyze the dustiness of powders, requires a minimum of six parallel samples due to its’ relatively poor repeatability [13]. And the drum exit would have to be modified/rewelded as it collects samples on dense, webbed metal plates, from which quartz cannot be analyzed. Similarly, pneumatic chambers used in some methods to prepare calibrators for quartz analyses cannot be used to estimate the portion of respirable dust as they do not yield reproducible portions of the respirable dust present in any given material [14,15]. Therefore, it cannot be used to reproducibly measure minute proportions of respirable quartz from powdery materials [15]. Consequently, in order to analyze the respirable quartz content of powdery products or materials, separation of the respirable fraction is often accomplished by sedimentation in liquid matrixes based on Stokes law [16]. According to Stokes law, the sedimentation velocity of a spherical particle is dependent on the squared hydrodynamic diameter of the particle (Equation (1)). From this, the size-weighed relevant fine fraction (SWeRF) of an aerosol can be derived by combining the particle size distribution of a powder with the probability factors from standard EN 481 [12], to enable the calculation of the relevant fine fraction of a material [16,17]. This approach is preferred for practical purposes, rather than making the powder airborne and using cyclones to separate the corresponding aerosol fraction. The calculations involved were first described by Pensis et al. (2014) [17] and were adapted in EN-17289-3:2020 [13]. In addition to respirable silica, the same methodology has been applied also to, for instance, testing bitumen and related roadmaking products [18].
(1)$$V_{s}=\frac{1}{18}\frac{\left(\rho_{p}-\rho_{f}\right)}{\eta }gD^{2}$$
where

*V_s_* = sedimentation velocity of particles (m/s)*D* = hydrodynamic diameter of particles, i.e., Stokes diameter*g* = gravitational field strength (m/s^2^)*ρ_p_* = mass density of particle (kg/m^3^)*ρ_f_* = mass density of the fluid (kg/m^3^)*η* = dynamic viscosity of fluid (kg/(ms))

Albeit sieves have been used for the same purpose [19], the sedimentation of powdery materials to yield the respirable fraction prior to quartz analysis is the preferred method for quartz analysis of materials [16]. For this purpose, measuring glasses have been used for sedimentation before collecting the supernatant under vacuum using a pipette or siphon tube held at the sedimentation height of particles not included in the SWeRF [17]. Another option is the Andreasen sedimentation apparatus (pipette), as described by Guldner et al. [20]. The SWeRF can also be collected by tapping it at the estimated sedimentation height using a measuring glass, operating on the same principle as the Andreasen pipette [18]. In the present method, we opted to use sedimentation flasks, as described by Öhman 1968 [21] (Figure 1). Öhman used flasks to extract fine particles corresponding to the Johannesburg convention (50% cut-off point at 5 µm) [22]. We, however, chose to estimate the SWeRF corresponding to the respirable fraction, as described in EN 481, similarly to Pensis et al. [16,17], in accordance with the methodology recommended in the NEPSI treaty. In developing the method, we were aiming at a limit of detection far below 1 wt.%, as recommended by the industrial mineral producers in the EU, and even clearly below the 0.1 wt.% stipulated in Regulation (EC) No 1272/2008 [8,10]. The reason for this is that, according to workplace measurements, levelling of ceilings with spray plaster, using products containing less than 1 wt.% of silica and ca. 0.1 wt.% of respirable silica, can lead to high exposure of workers (>0.1 mg/m^3^ during 8 h), provided large areas are covered [7].

## 2. Materials and Methods

### 2.1. Sedimentation Principle and Parameters

Equation (1) is valid for spherical particles in static fluids, where the sedimentation velocity is defined by the Stokes diameter, i.e., the diameter of a sphere that has the same density and settling velocity as the particle [23]. Quartz particles from cement and other powdery products are usually not round but, rather, rough and jagged, so the hydrodynamic diameter (Stokes diameter) depicted in the present paper is the diameter of a spherical particle that has the equivalent density and sedimentation velocity as the quartz particle [24].

In our experience, for the respirable fraction to be efficiently separated, the settling time of the particles must be at least 20 min. In addition, the supernatant containing the respirable fraction must be collected without allowing larger settled particles with a diameter > 10 µm to be collected. The sedimentation flask shown in Figure 1 can be used for this purpose, as demonstrated by Öhman [21]. Since the particles are originally suspended throughout the vessel, the possible sedimentation distance will be longer for particles originally close to the surface than for particles near the bottom of the flask. Therefore, the sedimentation procedure with accompanying retainment of the supernatant must be repeated at least three times. In our case, we opted to use six repetitions to obtain a better yield.

As described by Pensis et al. [17], the sedimentation time of the respirable fraction (SWeRF) in a fluid can be derived from Equation (1) (Equation (2)).
(2)$$t=h\frac{18\eta }{\left(\rho_{s}-\rho_{m}\right)g}\frac{4}{9}{\left(\frac{\sqrt{\frac{\rho_{s}}{\rho_{water}}}}{\sum_{Aero\;0\;µm}^{Aero\;10\;µm}\left(EN\;481\right)}\right)}^{2}$$
where

t = sedimentation time, s*h* = sedimentation height, m = 0.033 m in the sedimentation flasks (Figure 1)*η* * = viscosity of fluid, kg/ms*g* = 9.81, m/s^2^ (standard acceleration of gravity)ρ*_s_* ** = density of particles, kg/m^3^ρ*_water_* * = density of water, kg/m^3^ρ*_m_* * = density of sedimentation fluid, kg/m^3^$\sum_{Aero\;0\;µm}^{Aero\;10\;µm}\left(EN\;481\right)$ = the integral describing the hydrodynamic diameter of respirable particles (0–10 µm) = 4.2818 × 10^−6^ m (according to *EN* 481).* dependent upon temperature.** should be determined according to EN15051/Annex C [13].

Sedimentation in ethanol at a temperature of 22 °C using the sedimentation flask (Figure 1) yields a sedimentation distance of 0.033 m and a sedimentation time of 39.71 min for the respirable fraction:(3)$$t=0.033\frac{18\times 0.001138}{\left(2648-787.6\right)9.81}\frac{4}{9}{\left(\frac{\sqrt{\frac{2648}{997.802}}}{4.2818\times 10^{-6}}\right)}^{2}\left[s\right]\equiv 39.71\;\min\;$$

### 2.2. Sedimentation (Öhman Procedure) and Gravimetry

The material to be analyzed was oven dried for at least 1 h (105 °C, Verticell 55, MMM Medcenter, Berner Ltd., Helsinki, Finland) and allowed to cool in a desiccator for a minimum of 12 h. Parallel samples of 50–1000 mg were weighed as described in ISO 15767 [25], with a precision balance having a readability of 0.001 mg (Mettler Toledo XP56, Mettler-Toledo AG, Greifensee, Switzerland). Temperature was controlled during weighing and sedimentation (22° ± 0.5 °C) but the relative humidity was not. For each batch of samples (typically six/day), one control sample consisting of 500 µg of respirable quartz (Respirable Alpha Quartz, median particle size 3.3 µm, purity 96.73 ± 0.21%, NIST SRM 1878b, Merck kGaA, Darmstadt, Germany) was weighed to be prepared and analyzed identically to the samples. 

Upon weighing, the materials were suspended in 500 mL of ethanol (Spiritus Fortis A, 96%, Berner Ltd., Helsinki, Finland) and transferred to a sedimentation flask, shaken by hand, and left to sediment. When the sedimentation time (ca. 40 min, Equation (3)) had passed, the supernatant containing respirable particles above the given sedimentation height (0.033 m) was sucked into the collection vessel (Figure 1) with the help of a water pump. The suspended respirable particles were retained under suction on a pre-weighed silver membrane filter (SKC 225-1803, pore-size 0.8 µm, diameter 25 mm, SKC Inc., Eighty Four, PA, USA), allowed to dry at room temperature (22 °C), and weighed with the same precision balance to yield the weight percentage of respirable particles per dry weight. After gravimetry, the filter was analyzed by X-ray diffractometry, as previously described [6].

### 2.3. X-ray Diffractometry 

Analyses were performed using a PanAnalytical diffractometer (PanAnalytical X’Pert Pro PW 3040/60, 2012), as described previously [6]. The α-quartz diffraction lines 4.26 Å, 3.34 Å, and 1.82 Å appeared at 2 θ angles of 20.85°, 26.67°, and 50.15°, respectively. These were all used for qualitative verification, while the main peak at 26.67° was used for quantitative analysis. The quantitative limit of determination was 10 µg. Control samples and calibrators were treated and analyzed identically to actual samples. If the results of the control samples deviated from the added amount by more than ±30%, recalibration was executed. Results were calculated from the mean of two parallel samples.

### 2.4. Sedimentation According to Standard AS 2341.27 [18] and Pensis et al. [17]

Prior to choosing the sedimentation flasks used by Öhman (Figure 1), we tested the procedure described in the Australian standard AS 2341.27 [18], using 500 mL measuring glasses equipped with a tap at the sedimentation height and another at the top to allow for compensation of underpressure during collection of the supernatant (Figure 2). Subsequently, we tested normal 500 mL measuring glasses for the same purpose but collecting the supernatant at the sedimentation height using suction through a pipette, as described by Pensis et al. [17]. Briefly, in this method, dried, weighed parallel samples and controls were suspended in 500 mL of ethanol as described above and allowed to sediment once at 22 °C in the measuring glasses for 258 min (as calculated from Equation (2)), prior to collection of the supernatant using a water pump connected to a pipette placed at a sedimentation height of 0.215 m. After collection, the particles were retained on silver membrane filters prior to drying, weighing, and quartz analysis.

### 2.5. Determining the Linear Range, Repeatability, and Accuracy of Analysis

A calibration curve was prepared from respirable quartz using the primary diffraction peak of quartz at 26.67° (Standard Reference Material NIST SRM 1878a, Respirable Alpha Quartz, median particle size 1.6 µm, purity 93.7% ± 0.21%, Merck KGaA, Darmstadt, Germany). Firstly, a stock solution was prepared by adding 50 mg of the reference powder to 1 litre of ethanol. The purity of the reference powder was accounted for. Upon silting in ethanol, the solution was sonicated for 20 min (Ultra Sonic Cleaner USC-TH, Avantor Inc., Radnor, PA, USA), after which it was moved to a magnetic stirrer (IKA Vortex 2, IKA-Werke GmbH & Co., Staufen, Germany). From this stock solution, an eleven-point standard curve was prepared spanning from 10 µg (0.2 mL stock solution) to 5000 µg (100 mL stock solution) of quartz added per calibrator. The calibrators included one blank calibrator containing only pure ethanol and no quartz. Calibrators and a control sample included with each batch of analyses were treated identically to native samples. This included the same sedimentation procedure and sample collection as used with all samples.

Repeatability (% CV) of the method was calculated from six samples of cement, containing (on average) ca 0.034% of respirable quartz per dry weight, as well as from five samples of cement to which ca. 50 mg of reference quartz powder was added (see above), with the purity of the reference powder accounted for. Accuracy in addition to repeatability was calculated from six control samples containing ca. 500 µg (500–530 µg) of the NIST reference quartz powder, with the purity of the reference powder accounted for. Yield and accuracy were also counted from the aforementioned five cement samples to which a known amount of quartz was added. The yield was estimated from a standard curve drawn up without the sedimentation procedure. These calibrators were, thus, applied directly on silver membranes, without subjecting them to sedimentation. The yield of the sedimentation procedures by Pensis et al. [17] and AS 2341.27 [18] were calculated using these same calibrators from two samples prepared as described above, using and analyzed identically to other samples.

The limit of detection of the method based on the Öhman procedure was determined by dilution to be 10 µg. Below this, all three qualitative diffraction peaks required were not visible.

### 2.6. Visualization of Particle Sizes by Scanning Electron Microscopy (SEM)

Particles retained on polycarbonate filters (0.8 µm pore size, 25 mm diameter, Nuclepore straight through pore membrane filters, Merck KGaA, Darmstadt, Germany) were gilded prior to analysis using a Bal-Tec SCD 050 device (BalTec Maschinenbau AG, Pfäffikon, Switzerland). The Au-coated filters were analyzed by SEM, as described in ISO standard 14966 [26]. Briefly, ca. one-quarter portions of the filters were cut out and the sections analyzed using a JSM 6610 LA (JEOL Technics Ltd., Tokyo, Japan) scanning electron microscope. A magnification of 500 was used. Quartz particles were identified by obtaining energy-dispersive X-ray (EDX) spectra and comparing the Si/O ratio to reference spectra.

## 3. Results

The respirable fraction collected after six consecutive sedimentations using the method based on the Öhman method of sedimentation did not contain particles with a Stokes diameter larger than 10 µm (Figure 3). Consequently, the sedimentation matrix, the sedimentation time deployed, the shape of the sedimentation flask, as well as the means of supernatant collection were such that particles not belonging to the respirable fraction were excluded from the sample.

Using a sedimentation procedure based on AS 2341.27 [18], while yielding a high percentage of particles smaller than 10 µm in Stokes diameter, included particles with a larger diameter as well (Figure 4). Similarly, we were unable to set the collection velocity low enough to exclude particles with a diameter larger than 10 µm when testing the method by Pensis et al. Consequently, we opted to use the Öhman method of sedimentation for future needs.

The yield of the method based on Öhman was estimated to be 78%, calculated from cement to which pure reference powder was added (Figure 5). The comparable yield of the sedimentation procedure described by Pensis et al. was found to be 39%, while the yield of the procedure described in AS 2341.27 was 133%. From the same spiked samples, the repeatability of the Öhman procedure expressed as relative standard deviation was 11%. The comparable repeatability of this method was 21%, when calculated from samples prepared from quartz reference powder (Figure 6), and 34% when calculated from samples of pure cement (Figure 7). The respective accuracies of the method were 83% from spiked cement and 91% using pure reference powder.

The quantitative (linear) range of calibration was 10–5000 µg of quartz per sample. This corresponds to 0.02–10 wt.%, when analyzing 50 mg samples of powdery construction materials (Figure 8). The Pearson correlation coefficient of the calibration curve was 0.9982 (r^2^ = 0.9963).

## 4. Discussion

The present method based on the Öhman procedure [21] gave a better yield (78%) than the measuring glasses used by Pensis et al. (39%) [17], even though the sedimentation height in the Pensis method was considerably higher (0.215 m vs. 0.033 m). A higher yield was to be expected, since the sedimentation was repeated six times in the present method, and in the Pensis method only once. Some losses are associated with sedimentation, regardless of the method used, as particles will, to some extent, adhere to the surface of the flasks and the collection lines. Also, each sedimentation will leave a significant portion of respirable particles in the sedimentation flasks. Regardless of the yield being lower, however, the Pensis method is as repeatable as the present method. From samples of diatomaceus earth, Pensis et al. estimated the relative standard deviation to be 9% [17], while in the present study, the method based on Öhman yielded a relative standard deviation of 11% from samples of cement to which quartz was added.

The comparable yield of the method described in AS 2341.27 [18] was calculated to be as high as 133%. The yield was most likely high due to the taps withdrawing particles sedimented below the sedimentation height (Figure 4). Method AS 2341 was developed for more viscous products, such as bitumen, and seems to work better for the purpose it was intended.

The benefit of using the method presented here, when compared to the method by Pensis [17] or the adaptation of it according to EN-17289-3 [19], is that the yield of the sample preparation and, hence, the limit of quantitation are better. Also, the likelihood of including particles larger than 10 µm seems to be smaller. This has to do with the shape of the sedimentation flask and the height of the sedimented fraction in relation to the supernatant, as much as the laminar velocity of the supernatant in the pipette or siphon tube used to collect the supernatant. With the “Öhman” sedimentation flask, almost no particles with an aerodynamic diameter larger than 10 µm were collected from below the given sedimentation height and, therefore, it is unlikely that the results are overestimates. And since the yield of the sedimentation procedure was close to 80%, it is just as unlikely for the results to be underestimates.

Another major advantage with the present method is that the respirable fraction collected from the supernatant after sedimentation can be analyzed directly by XRD. In the other procedures mentioned above [16,17,18], the supernatant is collected in a beaker, evaporated on a hotplate, and the residue collected for XRD or FTIR analysis. This would necessitate transferring particles to FTIR tablets or XRD sample cups from the beaker. Alternatively, redissolution of the evaporation residue is required in, for instance, isopropanol, followed by collection of particles to silver membrane filters under vacuum, prior to analysis by XRD. The first route will most likely yield a higher limit of quantitation than with the present method, and the second option mentioned is likely to further lower the yield because of additional sample treatment steps.

The correlation of the calibration curve was well above 0.99, and larger standards could be included. But, as the range of percent dry weight needed to be analyzed was 0.1–10 wt.%, we opted to depict results higher than 10 wt.% as >10 wt.%. Percentages lower than 0.02% of respirable quartz per dry weight need not be analyzed. That would not affect the interpretation of results, since the limit of detection (10 µg/0.02 wt.%) is one-fifth of the limit given in EC regulation No 1272/2008 [8]. Similarly, the highest amount analyzed quantitatively (5000 µg/10 wt.%) corresponded to the upper limit in the recommendation given by industrial mineral producers in the EU [10].

## 5. Conclusions

The developed method, including sample treatment and analysis, is well suited for the purpose of analyzing powdery construction materials and other powdery samples containing 0.02 wt.% to 10 wt.% of respirable quartz or other crystalline silicas per dry weight. The method was tested with samples of commercial cement and commercial plasters, containing ca. 1 wt.% of respirable particles. Therefore, it is possible that in products containing a higher percentage of particles with a hydrodynamic diameter of smaller than 10 µm, with the quartz content simultaneously being close to 0.1 wt.% or less, the filters used to collect the dust will be clogged. With such samples, dilution of the sample or analyzing smaller samples down to 5 mg is possible. Smaller sample amounts than this would, however, raise the quantitative limit of detection above 0.1 wt.%, and, consequently, the analysis would no longer meet the current demands. Also, analyzing very small fractions of the material delivered to the laboratory may be problematic with respect to the representability of the samples, and it may also lower the yield, accuracy, and repeatability of the analyses. Moreover, it should be considered that the particles to be sedimented must be completely de-agglomerated and should be able to sediment freely, unhindered by other particles. Hence, the volume of the powder should be less than 1 wt.% of the sedimentation volume (100 mL) [11], yielding a maximum powder volume of 1 mL (<2.6 g).

The method is, similar to other methods based on sedimentation, prone to human error, the reason being that sedimentations and dosage of samples and calibrators involve a considerable amount of handwork. For these reasons, the repeatability from different matrixes varied from 11 to 34%. Therefore, it is advisable to use parallel sampling when analyzing samples and calibrating, either to exclude errors or to account for a less than perfect repeatability of the sample treatments, regardless of the method used.

## Figures and Tables

**Figure 1 toxics-12-00241-f001:**
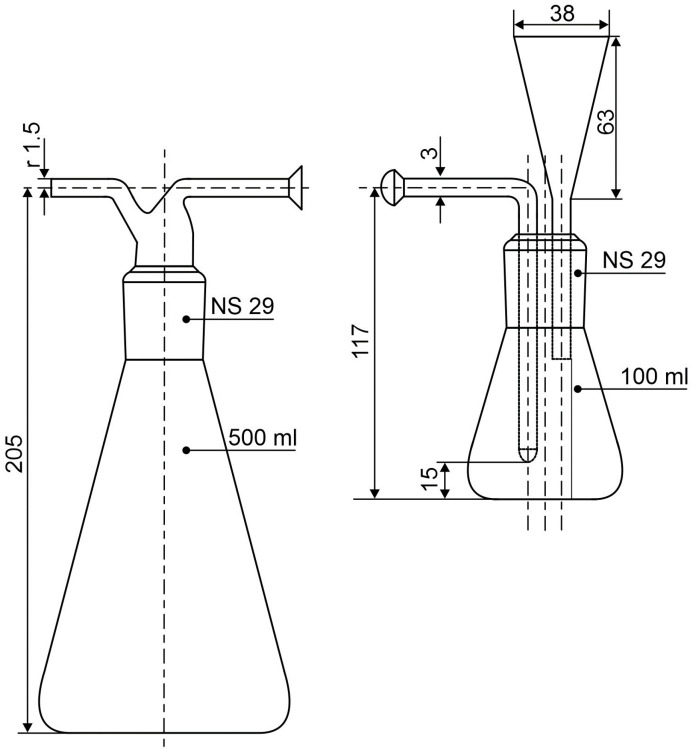
Liquid sedimentation flasks. Sedimentation was accomplished in the 100 mL Erlenmeyer flask on the right, and the SWeRF was collected in the 500 mL flask on the left (adapted from Öhman [21]).

**Figure 2 toxics-12-00241-f002:**
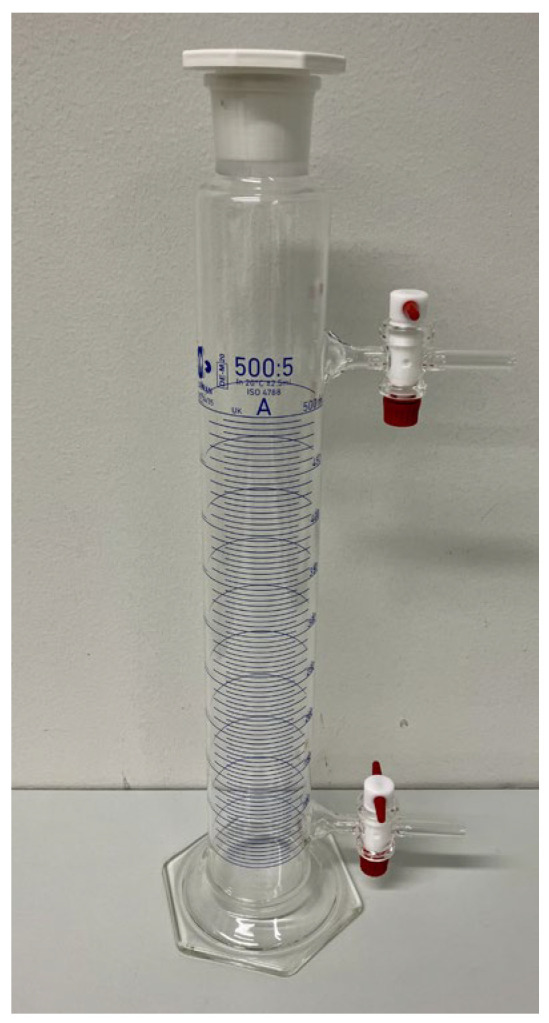
Measuring glass used in sedimentation as described in Australian standard AS 2341.27 [18].

**Figure 3 toxics-12-00241-f003:**
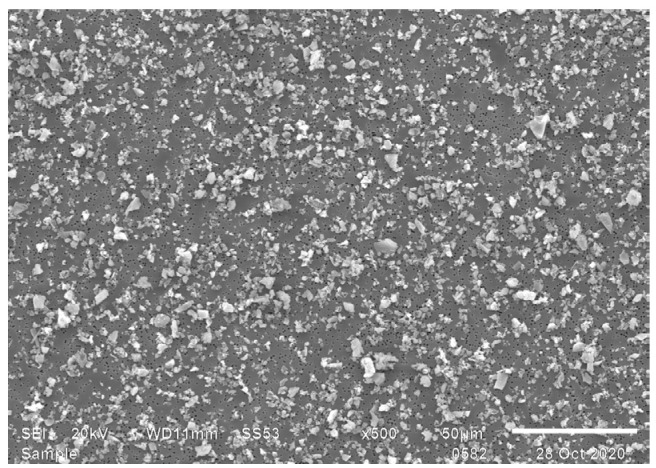
The respirable fractions combined from 6 consecutive sedimentations of cement in ethanol at 22 °C, using the Öhman sedimentation flask, with a sedimentation time of 40 min.

**Figure 4 toxics-12-00241-f004:**
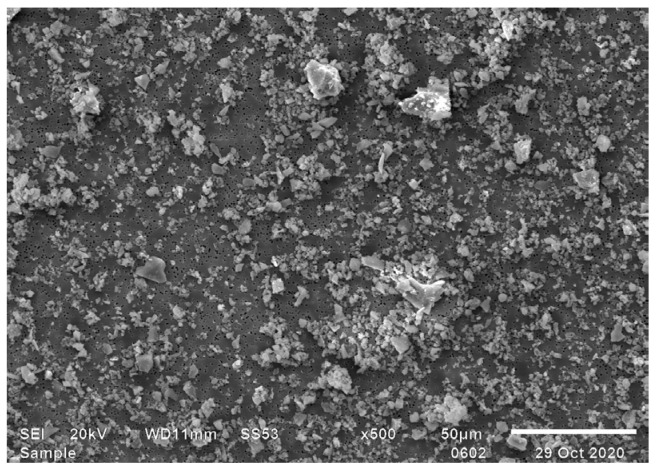
The respirable fraction from sedimentation according to AS 2341.27 [18].

**Figure 5 toxics-12-00241-f005:**
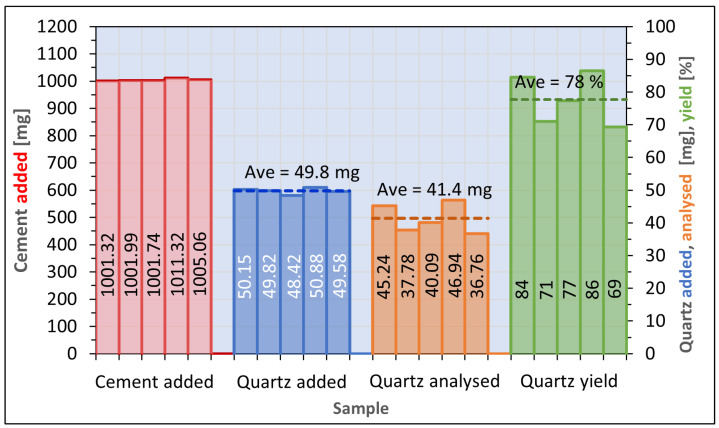
Results of test pertaining to repeatability, using cement to which quartz reference powder was added (Öhman method), quartz analysis repeatability (CV) 11% and accuracy 83%.

**Figure 6 toxics-12-00241-f006:**
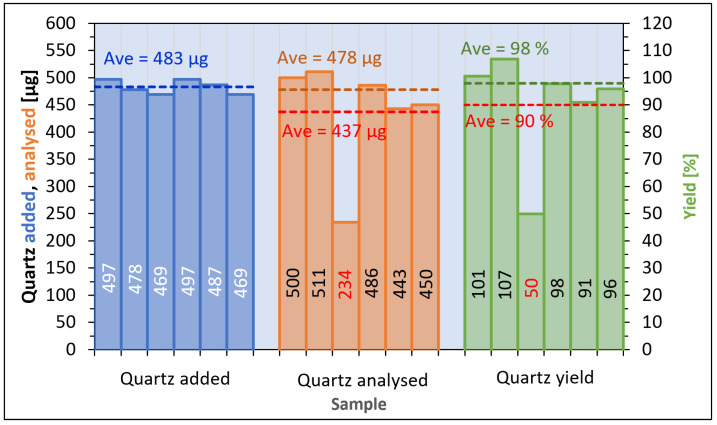
Results of test pertaining to accuracy and repeatability, using quartz reference powder (Öhman method), quartz analysis repeatability (CV) 21% and accuracy 91%. If sample 3 (red numbers) is ignored average of quartz sample analyzed is 478 µg, average quartz yield 98% and repeatability (CV) 6%.

**Figure 7 toxics-12-00241-f007:**
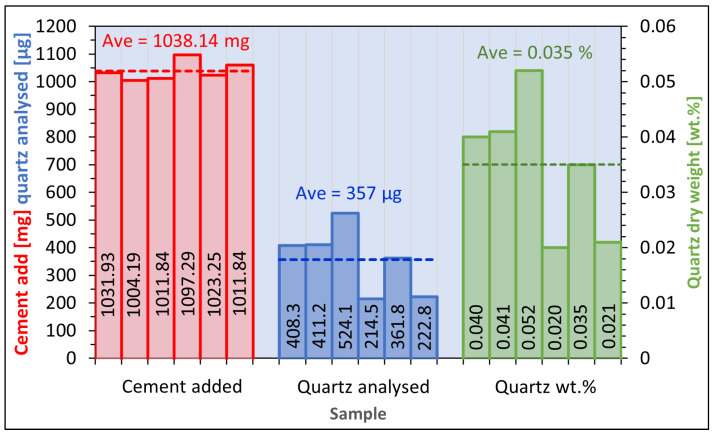
Results of test pertaining to repeatability, using cement with no quartz added (Öhman method), repeatability (CV) of quartz per dry sample is 21%.

**Figure 8 toxics-12-00241-f008:**
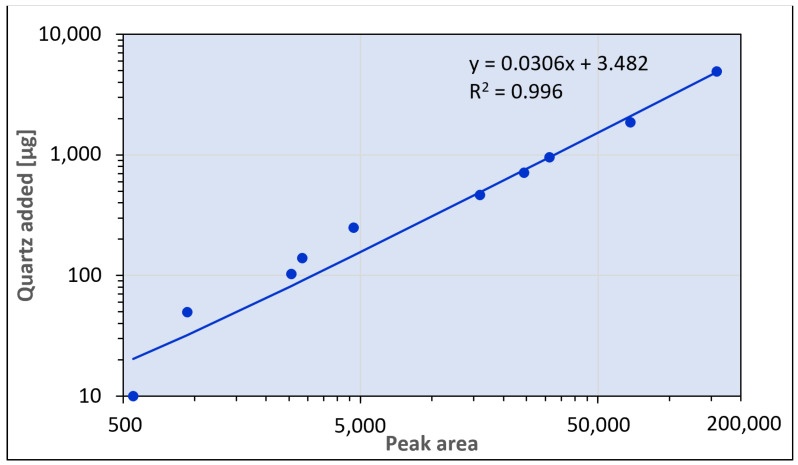
Calibration curve of the method (log. scale, 10–5000 µg quartz/calibrator).

## Data Availability

Raw data produced in the study are saved in the Laboratory Management System of the Finnish Institute of Occupational Health can be made available upon request from the institute. The data are not publicly available due to institute policy concerning information privacy.

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
