# Peer review of "Measuring Respirable Crystalline Silica (Quartz) from Powdery Materials through Sedimentation and X-ray Diffractometry"

_toxics, 2024, doi:10.3390/toxics12040241_

Round 1

Reviewer 1 Report

Comments and Suggestions for Authors

Introduction, the objective of the study should be clearly stated and benefit of the sedimentation method over the air sampling with respirable size-selective sampler should be also stated. The air sampling can be an easier method and does not have multiple steps of procedures compared to the liquid sedimentation procedures that is time consuming task (described in the manuscript); it should be discussed in the Discussion section rather than compared to the different sedimentation methods. It sounds like the study used the same method that was used by Ohman but it says “the developed method” and it is not clear. In addition, the yield rate of the respirable alpha quartz in the cement was not that high (<80%) so it should be discussed. The application of the method for the real samples is uncertain.

Line 93-95 According to EN 481, the percentage of the inhalable fraction which is to be included at any given aerodynamic diameter shall be given by named distribution and varies from 1.3 % at 10 μm (0 % at 16 μm) to 97 % at 1 μm, with a so called 50 % cut-off at 4 μm. It sounds like respirable fraction not inhalable fraction and the percent is sampling efficiencies (?). the percent presented here sounds like respirable fractions.

Line 282-283 not clear statement. What is “in any significant amount? How did the study determine the Stokes diameter? Typo anounts.

Line 295-296 “resulted in some particles with a larger diameter (Figure 4)” Any quantification rather than “some”? The Figure is only showing a small portion of the sample.

Author Response

Dear Reviewer 1,

Please find attached answers to the questions raised with references to the relevant lines that were rewritten.

Sincerely, Tapani Tuomi and Jussi Lyyränen

Reviewer 2 Report

Comments and Suggestions for Authors

I went through the manuscript entitled “Measuring respirable crystalline silica (quartz) from powdery 2 materials by sedimentation and x-ray diffractometry” by T. Tuomi and J. Lyyränen, finding it worth of consideration. The manuscript reports on an experimental procedure able to determine the presence of quartz in amounts that are in good relation to the limits imposed by law.

The article seems well organised, as well as the findings robust and enough documented. In the following, I list my remarks on the present version of the text, I recommend authors to take in due care before proceeding to finalise the revised version.

Major remarks:

1 – I partially disagree with the first statement of the abstract (“Choosing materials with a low quartz content is the most effective and cost-efficient way to prevent respirable quartz exposure of workers and other end users of powdery products”). This is true only in those application where replacement of silica-bearing materials is cost-effective. There are numerous examples in the case of construction work activities. I can cite the attempt to replace quartz sanding with garnet sanding. Enormous improvement in terms of occupational hygiene, but operations are too expensive and they are no more carried on.

2 – most of the analytical findings of your study rely in the experimental strategy at the X-ray diffractometer. As far as I can understand, you operate a quantification through a calibration procedure. There are some points that in my opinion should be better clarified:

-        How is physically prepared a calibrator?

-        There could be some problems due to the uneven distribution of particles when they are distributed over a surface (if the calibrator is a flat surface)?

-        Could you show the calibration curves (at least in a supporting information file)?

-        At the lines 216-218 you mention that you are considering 4 lines of the quartz powder pattern: how these lines are accounted for in the final regression curve? There are some differences among them?

3 – authors are surely aware of the fact that such information, at least for samples having > 1 wt% quartz, could be determined also by the Rietveld method. This procedure applies very easily on polycrystalline polyphasic materials (even respirable in size). Could you discuss the advantages of your method with respect to the Rietveld one in the Discussion paragraph? Could also in your opinion esteems on the rich side of your calibration curves be comparable with those obtained on the same materials through Rietveld refinement?

Minor remarks:

1 – the Stokes law equation, cited at line 101, is thus shown at line 111. Why not anticipating it the first time it is cited?

2 - why using silver membranes if you have prepared a calibration curve? Could you use the Silver diffraction peaks as internal reference?

3 - change “in any significant anounts” in “in any significant amounts” at line 284

Author Response

Dear Reviewer 2,

Please find attached answers to the questions raised with references to the relevant lines that were rewritten.

Sincerely, Tapani Tuomi and Jussi Lyyränen

Round 2

Reviewer 1 Report

Comments and Suggestions for Authors

Line 98-108 The chamber described in the MDHS 101 is using for generation of dust and air sampling is needed for respirable fraction samples. Is there any scientific publication related the statement “it cannot be used estimate the portion of respirable dust as the chamber does not yield reproducible portion of the respirable dust present in any given material”?

It might be better to cite studies that compared respirable fractions collected between cyclones vs. sedimentation in the discussion section.

Author Response

The statement sited in the reviewers comment pertaining to the chamber described in MDHS 101 (“it cannot be used estimate the portion of respirable dust as the chamber does not yield reproducible portion of the respirable dust present in any given material”) is based on our own experiences with using the chamber. This can also be deduced when considering the method description in MDHS 101 and the operating principle of the chamber. This is a direct quote from the method description (paragraph 54):

"Place about 0.1 g of the standard quartz or cristobalite in the bowl... Apply a jet of compressed air to the side arms of the bowl for a few seconds. Allow
approximately one minute for the coarser particles to settle out from the dust cloud. Run each sampling pump for sufficient time, typically 5-20 sec, to obtain filters loaded with the required amounts of standard covering the range 20-500 µg by trial and error. A minimum of five or six filters is required to give a reliable calibration"

As can be seen from the method description (paragraph 55), the calibration when using the chamber is based on weighing the filters from the collecting cyclones. The collection yields variable amounts of respirable quartz powder each time the collection is performed. Or more exactly, ca. 20-500 µg from a total sample of 0,1 g of quartz powder. The chamber works for this purpose when using pure quartz reference powder to yield calibrators with enough quartz for them to be weighed. But it is not possible to  generate a pulse of air that will give a reproducible amount of respirable dust on the filters. Each time a variable amount of respirable dust present in the initial sample will adsorb to the surfaces of the glass chamber. This is apparent, when looking at paragraph 55 in the MDHS 101 method description:

"When the required filters have been loaded with standard, allow them and the blanks to equilibrate in the balance room and re-weigh. Use the weights of the blank filters to determine a ‘blank correction’ for adsorption/desorption of  moisture and apply it to the weight increases of the loaded filters to obtain the mass of standard on each."

In other words, the chamber is well suited for the purpose it was intended, but not for analysing reproducibly the proportion of respirable dust or respirable silica in powdery materials with varying percentages of respirable dust and respirable crystalline silica. This being the case, we felt that it was enough to site the method in question (MDHS 101) in the previous version of the manuscript. As suggested, we did, however, ad one additional reference, that is a paper by Stacey et al. from 2019, where he states the same point as follows:

An evaluation of the method’s performance on replicate samples was not possible "because the amount loaded onto a filter from the generation of an aerosol of dust in a glass chamber cannot be accurately repeated."

See lines 106-108 and reference 15 in the new version of the manuscript.

Reviewer 2 Report

Comments and Suggestions for Authors

I appreciate the operated changes and the frank answer in the rebuttal letter. I guess after these changes the manuscript is improved. 

Author Response

We are much grateful and agree that the manuscript was improved.